# Effects of Hypoxemia by Acute High-Altitude Exposure on Human Intestinal Flora and Metabolism

**DOI:** 10.3390/microorganisms11092284

**Published:** 2023-09-11

**Authors:** Ping Qi, Jin Lv, Liu-Hui Bai, Xiang-Dong Yan, Lei Zhang

**Affiliations:** 1The First Clinical Medical College, Lanzhou University, Lanzhou 730000, China; qip21@lzu.edu.cn (P.Q.); lvj21@lzu.edu.cn (J.L.); bailh21@lzu.edu.cn (L.-H.B.); yanxd21@lzu.edu.cn (X.-D.Y.); 2Department of General Surgery, The First Hospital of Lanzhou University, Lanzhou 730000, China; 3Key Laboratory of Biotherapy and Regenerative Medicine of Gansu Province, The First Hospital of Lanzhou University, Lanzhou 730000, China

**Keywords:** acute high-altitude hypoxia, hypoxemia, gut microbiome, metabolism, erythropoietin

## Abstract

This study examined the effects of hypoxemia caused by acute high-altitude hypoxia (AHAH) exposure on the human intestinal flora and its metabolites. The changes in the intestinal flora, metabolism, and erythropoietin content in the AHAH population under altitude hypoxia conditions were comprehensively analyzed using 16S rRNA sequencing, metabonomics, and erythropoietin content. The results showed that compared with those in the control group (C group), the flora and metabolites in the hypoxemia group (D group) were altered. We found alterations in the flora according to the metabolic marker tyrosine through random forest and ROC analyses. Fecal and serum metabonomics analyses revealed that microbial metabolites could be absorbed into the blood and participate in human metabolism. Finally, a significant correlation between tyrosine and erythropoietin (EPO) content was found, which shows that human intestinal flora and its metabolites can help to confront altitude stress by regulating EPO levels. Our findings provide new insights into the adaptive mechanism and prevention of AHAH.

## 1. Introduction

Plateaus are unique ecosystems characterized by low oxygen, cold temperatures, and high radiation levels, which have direct and lasting effects on human health [1]. Hypoxemia caused by acute exposure at high altitudes is one of the most common health problems in such conditions [2,3]. Hypoxemia caused by acute altitude exposure refers to the lack of oxygen supply due to the decrease in oxygen partial pressure when a person enters a high-altitude area from the plain or rises from a low- to a high-altitude area in a short time. Its main feature is that the oxygen saturation is lower than the normal range [4]. Its symptoms can be mild (e.g., headaches, nausea, vomiting, and chest tightness) or severe (e.g., pulmonary and cerebral edema) [5,6]. The harmful effects of hypoxemia on human health cannot be overlooked.

The gut microbiota is often referred to as the “invisible organ” of the body [7]. Its structural and functional integrity is essential for maintaining the intestinal barrier, assisting digestion and absorption of nutrients, and maintaining the immune system and metabolic homeostasis [8]. High altitudes and low oxygen supply to the human body can cause intestinal mucosal ischemia and damage to the intestinal barrier, leading to dysbiosis of the gut microbiota [9], which is associated with multiple diseases [10,11].

Previous studies have analyzed the human gut microbiome in high-altitude environments [12]; however, no one has explored how hypoxemia affects the composition and metabolism of intestinal microflora and their interaction with human metabolism. Therefore, a comprehensive and in-depth understanding of the effects of a high-altitude environment on human intestinal microorganisms and their metabolism is essential for solving human health problems in such conditions [13].

This study examined how hypoxemia caused by acute high-altitude exposure affects the composition and function of the gut microbiota and human metabolism and investigated the relationship between intestinal microflora imbalance, human metabolism, and EPO content. This study provides new insights into the prevention and treatment of plateau diseases.

## 2. Materials and Methods

### 2.1. Subjects

Gannan is located in the southwest of China, on the northeast edge of the Qinghai-Tibet Plateau and to the south of the Loess Plateau, with an average elevation of over 3000 m, belonging to a high-altitude area. In contrast, Lanzhou, located in Gansu Province, China, has an average elevation of only 1500 m. When people enter Gannan from Lanzhou, which is at a lower altitude, the altitude rises by 1500 m, which leads to acute high-altitude exposure, which causes some people to have symptoms of hypoxemia. The main feature of hypoxemia is that the oxygen saturation is lower than the normal range (95–100%) after acute high-altitude exposure [4]. Hypoxemia typically manifests within 6 to 12 h after exposure, reaching its peak severity after the initial night [14]. In the absence of additional altitude gain, hypoxemia generally improves within 48 h [15,16]. Therefore, when investigating the influence of hypoxemia on the intestinal flora, it is important to select individuals who experience hypoxemia within 48 h of entering high-altitude areas. Selecting a timeframe exceeding 48 h would likely involve individuals who have entered the high-altitude exposure adaptation phase, during which hypoxemia symptoms disappear for most patients.

To study this phenomenon, the researchers recruited 20 Lanzhou residents at Lanzhou University who developed symptoms of hypoxemia less than 48 h after entering Gannan. The researchers analyzed the microbial community and metabolic changes of these people when they first entered the Gannan area. The inclusion criteria for the participants were as follows: (1) no history of severe digestive system diseases; (2) no recent digestive discomfort or illness; (3) no other recent diseases or infections; (4) no recent use of medication; and (5) not pregnant, breastfeeding, or menstruating. The exclusion criteria were as follows: (1) clinical diagnosis of major intestinal diseases or chronic inflammation; (2) use of aspirin, insulin, metformin, statins, or metoprolol; (3) use of antibiotics or probiotics in the past 8 weeks; and (4) refusal to sign an informed consent form. Appendix A provides baseline data for 20 volunteers. This study was approved by the Ethics Committee of the First Hospital of Lanzhou University (approval number: LDYYLL2019-36), and informed consent was obtained from all participants.

### 2.2. Blood SAO_2_ Measurements

A Covidien Nellcor OxiMax N-600x pulse oximeter was used to measure blood SAO_2_. The sensor was clipped onto the fingertip of the subject to ensure its tight attachment and placed next to the subject. The measurement button was pressed to measure the blood SAO_2_, which was displayed on the screen. The pulse oximeter was used to measure the blood oxygen saturation of 20 volunteers before and after entering the Gannan area.

### 2.3. Sample Collection, Processing, and 16S rRNA Analysis

A total of 20 serum samples (fasting for 8 h) and 20 fecal samples were collected from Han Chinese individuals who had been residing in the Plains region for an extended period. Additionally, 20 serum samples (fasting for 8 h) and 20 fecal samples were collected from the same individuals after their first entry into the high-altitude plateau region.

After collecting 8 mL of blood from the study participants, serum samples were extracted by centrifugation at 22 °C for 10 min at 3000 rpm. The serum was then subpackaged, with 1 mL reserved for metabonomics measurement and 2 mL reserved for EP0 content determination. The serum and fresh fecal samples were promptly placed in an icebox and transported to the laboratory for processing. Tubes were prepared by accurately weighing 200 mg of fecal or 1 mL of serum samples and placing them into a sterile 2 mL centrifuge tube. The prepared samples were rapidly transferred to a −80 °C freezer for storage. The collection and packaging processes were completed within 30 min.

Microbial DNA was extracted from 40 frozen stool samples using the QIAamp DNA Stool Mini Kit (Qiagen, Hilden, Germany). The V3–V4 hypervariable region of the 16S rRNA gene was amplified using the ABI 2720 Thermal Cycler (Thermo Fisher Scientific, Waltham, MA, USA) with primers containing Illumina adapter sequences. The forward primer was the Illumina adapter sequence 1 + CCTACGGGNGGCWGCAG, while the reverse primer was the Illumina adapter sequence 2 + GACTACHVGGGTATCTAATCC. The PCR reaction included 10 ng of template DNA, 2 μL of 2.5 mM dNTP, 0.8 μL of each primer (5 μM), 0.4 μL of FastPfu polymerase, and 20 μL of 4 × FastPfu buffer; PCRs were performed in triplicate. The PCR products were extracted from a 2% agarose gel, purified using the AxyPrep DNA Gel Extraction Kit (Axygen Biosciences, Union City, CA, USA), and quantified using the Invitrogen Qubit3.0 Spectrophotometer (Thermo Fisher Scientific, USA). The purified amplicons were pooled in equimolar concentrations and subjected to paired-end sequencing on the Illumina NovaSeq Benchtop Sequencer platform (Illumina, San Diego, CA, USA) according to the standard protocol of Tianhao Biology Science and Technology Co. (Shanghai, China). Subsequent bioinformatics analysis was performed.

### 2.4. Untargeted Serum Metabolomics Analysis

Fecal samples were thawed at 25 °C for metabolite extraction. Here, 50 ± 1 mg of the sample was transferred into a 2 mL tube, and 1 mL of a pre-chilled extraction buffer (methanol/chloroform (*v*:*v*) = 3:1) containing 5 µg/mL adonitol was added. The samples were vortexed for 30 s and homogenized in a ball mill for 4 min at 45 Hz, followed by ultrasonication for 5 min in ice water. This process was repeated three times. After centrifugation at 4 °C for 15 min at 12,000 rpm, 150 μL of the supernatant was transferred to a fresh tube. For quality control (QC) samples, 100 μL of each sample was removed and combined. After removing the solvent using a vacuum concentrator, 30 μL of methoxyamination hydrochloride (20 mg/mL in pyridine) was added, and the samples were incubated at 80 °C for 30 min. Next, the samples were derivatized by adding 40 μL of the N,O-Bis(trimethylsilyl) trifluoroacetamide reagent (1% (*v*/*v*) trimethylchlorosilane) and incubated at 70 °C for 1.5 h. The samples were gradually cooled to 25 °C, and 5 μL of fatty acid methyl esters (dissolved in chloroform) was added to the QC sample. Finally, all samples were analyzed using gas chromatography coupled with time-of-flight mass spectrometry (GC-TOF-MS).

An Agilent 7890 gas chromatograph (Agilent Technologies, Santa Clara, CA, USA) coupled with a Bruker UltrafleXtreme MALDI-TOF/TOF mass spectrometer (Bruker Daltonics, Bremen, Germany) was used for GC-TOF-MS. The system was equipped with a DB-5MS capillary column (30 m × 250 μm × 0.25 μm; J&W Scientific, Folsom, CA, USA). A 1 μL aliquot of the sample was injected in the splitless mode. Helium was used as the carrier gas, and the front inlet purge flow rate was set to 3 mL/min. The gas flow rate through the column was set at 1 mL/min. The initial temperature was maintained at 50 °C for 1 min and then increased to 310 °C at a rate of 10 °C/min. The temperature was held at 310 °C for 8 min. The injection, transfer line, and ion source temperatures were set at 280, 280, and 250 °C, respectively. The energy used was −70 eV in the electron impact mode. Full-scan mass spectrometry data were acquired at a rate of 12.5 spectra per second in the *m*/*z* range of 50–500 after a solvent delay of 6.27 min. The Mass Spectrometry-Data Independent Analysis software version 4 and the Fiehn Binbase database [17] were used for raw peak extraction, baseline filtering, baseline calibration, peak alignment, deconvolution analysis, peak identification, and peak area integration [18]. Metabolite identification was achieved through mass spectrum matching and retention index matching. To ensure accuracy, peaks that were detected in less than 50% of the QC samples or exhibited a relative standard deviation greater than 30% in the QC samples were subsequently removed [19].

### 2.5. Human EPO Content Measurements

The Elabscience^®^ Human EPO ELISA Kit (product number: E-EL-H3640c) from Elabscience Biotechnology Co., Ltd. (Houston, TX, USA) was used to measure serum EPO content.

This assay kit utilizes a sandwich enzyme-linked immunosorbent assay (ELISA) method with dual antibodies. An anti-human EPO antibody is coated onto the plate, and during the assay, human EPO in the sample or standard solution binds to the coated antibody, while unbound components are washed away. Subsequently, a biotinylated anti-human EPO antibody and horseradish peroxidase-labeled avidin are added sequentially. The anti-human EPO antibody binds to the human EPO bound to the coated antibody, and the biotin binds specifically to the avidin. This results in the formation of an immune complex that is washed to remove any unbound components. A chromogenic substrate (TMB) is added, and under the catalysis of horseradish peroxidase, TMB turns blue, which subsequently turns yellow after the stop solution is added. The optical density (OD) is measured at 450 nm using an ELISA reader, and the EPO concentration is proportional to the OD450 value. The EPO concentration in the sample can be calculated by drawing a standard curve.

### 2.6. Statistical Analyses

Microbiome analysis: We performed 16S data analysis using R Studio 4.2.3. Specifically, amplicon sequence variant (ASV) clustering in ASV classification analysis was conducted using QIIME v2. Species annotation was also conducted using QIIME2. To group and rarefy ASVs, we utilized the R package phyloseq v1.26.1 and calculated species abundance and composition information based on the ASV abundance data. For alpha diversity analysis, we used the vegan package (v2.5.6) in R to calculate diversity indices, including Observed, Chao1, ACE, Shannon, Simpson, and Coverage indices. Additionally, R v3.5.1 was utilized for statistical analysis of the diversity indices. Dilution and Shannon–Wiener curves were generated using QIIME v2 software.

Principal coordinate analysis (PCoA) was utilized to explore differences in community composition between samples, and the R packages scatterplot3d v0.3.41, pheatmap v1.0.12, and phyloseq v1.26.1 were used for comparison. Furthermore, partial least squares–discriminant analysis (PLS-DA) was employed to investigate community composition differences between samples, utilizing the R package mixOmics v6.6.2. Lastly, the linear discriminant analysis effect size (Lefse) was determined using MicrobiomeAnalyst 2.0 to identify differential bacteria at the genus level between the control (C) group and hypoxemia (D) group.

Metabolome analysis: We conducted metabolomics analysis using MetaboAnalyst 5.0. Specifically, we employed a volcano plot to differentiate overall differences in metabolite spectra between C and D groups and used the fold change and *t*-test to select differentially abundant metabolites with the greatest contribution. These differentially abundant metabolites were then subjected to pathway enrichment analysis based on the KEGG database.

EPO data analysis: In the analysis of EPO data, we use the independent sample t-test analysis method in SPSSPRO to compare the differences in EPO levels between the two groups. Correlation analysis: To explore the relationship between the microbiome and metabolome, a correlation analysis was conducted using the corrplot package in R. A total of 20 participants who provided fecal and blood samples before and after entering the plateau were included in the analysis. Spearman correlations were calculated among the microbiome, metabolome, and EPO variables. A correlation heatmap was generated using the corrplot package to display the relationships among the variables.

Random forest and receiver operating curve analyses (ROC): We conducted these analyses using MetaboAnalyst 5.0. Random forest analysis has good classification prediction ability and different visualization ability, which can be used to understand the importance of variables and analyze potential biomarkers. ROC analysis is a comprehensive index reflecting sensitivity and specificity, which can be used for continuous variables. This analysis calculates a series of sensitivity and specificity parameters by setting critical values for continuous variables and then draws a curve with sensitivity as the ordinate and specificity as the abscissa. The larger the area under the curve, the higher the diagnostic accuracy. In this study, random forest analysis was used to explore the difference between D and C groups, and then the ROC curve was used to evaluate the accuracy of the screened biomarkers.

## 3. Results

### 3.1. 16S rRNA Findings

Based on the AVSs analysis, both the Shannon–Wiener and Sparse curves suggest that the sequencing depth achieved adequate coverage for investigating the gut microbiota of both groups. The Shannon–Wiener curve’s flattening (Appendix A) suggests that the sequencing data accurately represent microbial information in most samples. Similarly, the rarefaction curve’s flattening with an increase in extracted sequences (Appendix A) indicates that the sequencing data for the samples are reasonable. Appendix A displays the results for the AVSs of the C and D groups.

PCoA is a visualization technique that represents the similarity or dissimilarity of research data through coordinate visualization. It is a non-constrained method of dimensionality reduction analysis, commonly employed to investigate the similarity or dissimilarity of sample community compositions. The results of the PCoA analysis, based on the Bray–Curtis distance, reveal clear differences in the microbial communities between the CD groups (Figure 1a). The intestinal flora of the C and D groups were analyzed using PLS-DA, and the results showed that there were clear differences in the flora structure of both groups (Figure 1b), which indicated that the flora of the D group population was disordered. Differences in alpha diversity indices were analyzed using the Wilcoxon rank sum test, with a significance threshold set at *p* < 0.05. The results revealed no significant differences in the Observed, Chao, ACE, Simpson, and Coverage indices between groups C and D (Figure 2). However, the Shannon indices demonstrated a significant difference in microbial diversity between the two groups. Specifically, the flora richness of group D was greater than that of group C, suggesting an increase in the flora richness of the AHAH population. Appendix A displays the results for the alpha diversity indices of the C and D groups.

Only species with a relative abundance greater than 1% in each sample were selected to create a bar chart representing phylum and genus levels. This allowed for easy observation of the species composition of each sample, the consistency of species composition within each group, and differences in species composition between the groups. The bar chart revealed that at the phylum level, Firmicutes, Bacteroidetes, Actinobacteria, and Proteobacteria were the primary phyla in both groups (Figure 3a). Compared with the C group, the relative abundance of Firmicutes, Actinobacteria, and Proteobacteria was lower in DG, whereas the relative abundance of Bacteroidetes was higher in the D group. This suggests that the gut microbiota at the phylum level undergoes significant changes in response to acute high-altitude hypoxia (AHAH).

At the genus level, the bar chart indicated that the gut microbiota composition in the D group differed significantly from that in the C group (Figure 3b). The genera with decreased relative abundance in the D group included *Blautia*, *Lachnospiracea_incertae_sedis*, and *Bifidobacterium*, which are associated with short-chain fatty acid (SCFA) and carbohydrate metabolism. In addition, the relative abundance of pathogenic genera (e.g., *Escherichia* and *Shigella*) was decreased in the D group. The genera with increased relative abundance in the C group included *Bacteroides*, *Faecalibacterium*, and *Alistipes*, which produce SCFAs. Appendix A presents the results of our measurements of the abundance of the 16S phylum and genus taxa in the C and D groups. 

Lefse (linear discriminant analysis effect size), that is, LDA Effect Size analysis, is an analytical tool for discovering and interpreting biomarkers (taxonomic units, pathways, and genes) of high-latitude data, which can perform comparisons between two or more groups to find species with significant differences in abundance between groups. At the genus level, with the absolute value of the LDA score > 2.5 and *p*-value < 0.05 as the threshold, nine species of bacteria with differences among C and D groups were screened out (Figure 4), including *Odoribacter*, *Weissella*, *Saccharibacteria_genera_incertae_sedis*, *Enterobacter*, *Gemella*, *Bacteroides*, *Parabacteroides*, *Alistipes,* and *Akkermansia*. Of these, the abundance of *Odoribacter*, *Bacteroides*, *Parabacteroides*, *Alistipes,* and *Akkermansia* increased in the D group, whereas the abundance of *Weissella*, *Saccharibacteria_genera_incertae_sedis*, *Enterobacter*, and *Gemella* decreased in the D group.

### 3.2. Metabolic Findings in Stool Samples

Stool samples were subjected to GC-MS-based metabolomic analyses, revealing metabolic differences between groups C and D. A volcano plot was generated to screen for differential biochemical indicators between C and D groups, with the degree of difference measured by a fold change (FC) > 2 and a *p*-value < 0.05, where seven differentially accumulated metabolites were identified (Figure 5), including tyrosine, nonadecanoic acid, chlorogenic acid 1, chenodeoxycholic acid, and n-acetylglutamate, which were increased in the D group, while alpha-aminoadipic acid and 5-aminovaleric acid were decreased in D group. Appendix A displays the results of our measurements of all fecal metabolites in samples from the C and D groups.

### 3.3. Correlation Analysis between Differential Bacterial Genera and Differential Metabolites in Feces and Importance Screening with Random Forest Analyses

To determine the different fecal metabolites directly related to the flora disorder, we analyzed the Spearman correlation between seven identified metabolites and nine different bacteria. Taking the *p*-value < 0.05 as the significance threshold, the correlation heat map (Figure 6a) shows that among these metabolites and bacteria, there are significant differences between four metabolites and seven bacteria. These differential metabolites were identified as potential key differential metabolites related to flora disorder caused by AHAH.

Because of the high complexity of metabonomics data, it is imperative to adopt effective data analysis and pattern recognition methods for metabonomics data analysis [20]. Random forest is an effective method that explores the differences between metabolites in samples and provides the interaction between variables, which helps find potential biomarkers [21]. In this study, we used the random forest method to analyze the importance of different variables of four metabolites significantly related to the genus (Figure 6b). We finally determined that these four different metabolites were highly differentiated between the two groups, among which tyrosine was the most significantly different metabolite between the two groups.

These four metabolites are essential in various physiological processes, and the intestinal microflora significantly affects their contents. Tyrosine is an amino acid and a precursor of catecholamine, which can affect the synthesis of dopamine and norepinephrine [22]. The primary function of tyrosine is to improve cognitive function, and it can prevent its decline associated with stress in the short term, such as cold and high-altitude stresses (that is, mild hypoxia stress) [23,24]. In addition, the correlation analysis showed that tyrosine is positively associated with *Bacteroides*, *Parabacteroides*, and *Odoribacter*, while it is negatively correlated with *Saccharibacteria_genera_incertae_sedis*. N-acetylglutamate is a unique cofactor that plays an essential role in urea production in mammals. In addition, N-acetyl glutamate is the first common substrate for de novo arginine synthesis in microorganisms and plants [25]. The correlation analysis showed that N-acetyl glutamate is positively associated with *Bacteroides*, *Alistipes,* and *Parabacteroides*, while it is negatively correlated with *Gemella*. 5-aminovaleric acid is a δ amino acid, which is a methylene homolog of γ-aminobutyric acid and a weak γ-aminobutyric acid agonist [26]. The intestinal microbiota plays a critical role in the production of 5-aminovaleric acid [27]. The correlation analysis showed that 5-aminovaleric acid is negatively correlated with *Bacteroides*, *Alistipes*, *Parabacteroides*, *Akkermansia,* and *Odoribacter*, while it is positively associated with *Saccharibacteria_genera_incertae_sedis*. Chenodeoxycholic acid is the primary bile acid produced by cholesterol in the liver and one of the main bile acids in the human body [28]. Intestinal microorganisms can metabolize chenodeoxycholic acid and cholic acid through 7α-dehydroxylation, thus producing dehydrocholic acid and dehydrocholic chenodeoxycholic acid [29]. The correlation analysis showed that chenodeoxycholic acid is positively associated with *Odoribacter*.

### 3.4. Biomarker Analysis

The receiver operating characteristic curve (ROC) can be used to judge whether a particular factor has diagnostic value for a disease. The AUC (area under the ROC curve) indicates the prediction accuracy, with higher AUC values indicating greater accuracy. Thus, when the curve is closer to the upper left corner, the prediction accuracy is high (a small value of x corresponds to a large y value). The four potential biomarkers screened in Section 3.6 were analyzed by using the ROC, and the results are shown in Figure 7 and Table 1. The features displayed in the table below are ranked based on the AUC, t-statistics, and fold change (FC). Among the potential biomarkers, tyrosine exhibited the highest AUC value of 0.878 (with a sensitivity of 0.9 and specificity of 0.8), as illustrated in Figure 7a. These findings indicate that tyrosine has a high diagnostic accuracy for flora imbalance in acute high-altitude hypoxemia and is of significant clinical importance for early diagnosis. Additionally, the box plot in Figure 7b demonstrates a higher concentration of tyrosine in the D group compared to the C group.

### 3.5. Metabolic Findings in Serum Samples

Serum samples were subjected to GC-MS-based metabolomic analyses, revealing metabolic distinctions between groups C and D. A volcano plot was generated to screen for differential biochemical indicators between groups D and C, with the degree of difference measured by a log2 fold change (FC) > 2 and a *p*-value < 0.05. Eight differentially accumulated metabolites were identified, including tyrosine, nonadecanoic acid, n-acetylglutamate, chlorogenic acid 1, and chenodeoxycholic acid, which were increased in group D, while alpha-aminoadipic acid, 5-aminovaleric acid, and glycolic acid were decreased in group D (Figure 8). Appendix A displays the results of the measurements of all serum metabolites in samples from the C and D groups.

### 3.6. Correlation Analysis of the Same Differential Metabolites in Serum and Feces

Spearman correlation and significance analyses were performed on the differential metabolites in the feces and serum related to the dysbiotic microbiota (Figure 9). The correlation coefficients of the four metabolites screened from feces and blood were greater than 0.99, *p* < 0.01, showing a significant correlation, including n-acetyl glutamate, 5-amino acid, tyrosine, and chenodeoxycholic acid. This indicates that flora disorder leads to an increase in the content of four metabolites, which are absorbed into the bloodstream, increasing these metabolites in serum.

### 3.7. Pathway Enrichment Analysis 

Based on the screened metabolites and their contents in serum, a pathway enrichment analysis was performed using the KEGG database for the four metabolites, finding that the metabolic pathways of the differential metabolites were enriched. Pathways with a pathway impact > 0 and *p* < 0.05 (top 25) were selected, revealing significant changes in ubiquinone and other terpenoid-quinone biosynthesis, tyrosine metabolism, phenylalanine metabolism, phenylalanine, tyrosine, aminoacyl-tRNA biosynthesis, primary bile acid biosynthesis, and arginine biosynthesis, as shown in Figure 10.

### 3.8. EPO Content Determination

EPO is a glycoprotein hormone mainly secreted by the kidney that promotes the production of red blood cells, thus increasing the oxygen content in the blood [30]. During hypoxia, the body will produce more EPO to stimulate the bone marrow to produce more red blood cells.

As shown in Table 2, the average values of the EPO content of the C and D groups were 9.798 and 19.575 mIU/mL, respectively, finding a significant difference between both groups. Cohen’s d value of the difference in range was 1.022, and the difference in range was considerable, with 0.20, 0.50, and 0.80 corresponding to small, medium, and large critical points, respectively. This was consistent with previous research results [31]. These results show that in an AHAH environment, AHAH patients can produce more red blood cells by increasing EPO content to help the body cope with the stress of high-altitude hypoxia. The Appendix A shows the EPO content in samples from the CD group.

### 3.9. Correlation Analysis between EPO Content and Screened Serum Differential Metabolites

The Spearman correlation and significance between the four serum differential metabolites and EPO were analyzed (Figure 11). We found that there was a significant positive correlation between tyrosine and EPO levels. The correlation coefficient was 0.57 and *p* = 0.000141, indicating that during AHAH, flora imbalance could increase tyrosine content in the body, thus increasing EPO levels in the human body.

## 4. Discussion

This study aimed to investigate the changes in the intestinal flora and metabolites in subjects with AHAH and their effects on the human body. Here, we combined 16S rRNA, metabonomics, and EPO analyses.

To the best of our knowledge, this is the first study to reveal a correlation between AHAH and gut microbiota. Here, we first used oxygen saturation as a screening standard to remove subjects with AHAH by adopting a self-control method to reduce the influence of other factors, such as diet and smoking, on the microbial population [32]. Then, we further analyzed the microbial population of AHAH subjects and their metabolic changes, from the intestinal flora to human physiology, by analyzing the relationship between its differential metabolites, human metabolic pathways, and EPO.

The intestinal flora of AHAH patients changed significantly. The bacteria producing short-chain fatty acids changed, and the relative abundance of pathogenic bacteria decreased. In addition, the fecal metabolites also changed significantly. There were clear changes in nine different bacteria genera and seven metabolites in the feces of the AHAH subjects. Through the correlation analysis between different bacteria and different fecal metabolites, seven differential bacteria and four metabolites were found, and the random forest model analysis confirmed that the four metabolites were highly differentiated between the two groups and were identified as potential biomarkers. Then, the biomarkers of these four different metabolites were analyzed by using the ROC, and tyrosine was finally determined as the fecal metabolic marker of bacterial flora disorder in AHAH subjects.

Through screening serum differential metabolites in AHAH subjects, eight types of differential metabolites were found, and the correlation between these eight types of serum metabolites and four types of fecal metabolites was analyzed. The results revealed a significant and strong correlation between the contents of the four fecal metabolites and their serum counterparts (correlation coefficient > 0.99, *p* < 0.01), indicating that these fecal metabolites are absorbed through the intestine and then play a role in human metabolism. Pathway enrichment analysis of these metabolites found seven metabolic pathways with significant differences. Based on these results, we found that the disordered flora of the AHAH subjects can regulate human metabolism through its metabolites.

The most significant metabolic pathway identified in this study is ubiquinone and other terpenoid-quinone biosynthesis. This pathway involves the biosynthesis of ubiquinone, also known as coenzyme Q10, a vital terpenoid quinone antioxidant in the human body. Its functions in the human body include participating in the electron transport chain of mitochondria to promote ATP synthesis, protecting cells from oxidative damage caused by free radicals, and regulating cell apoptosis and the cell cycle [31,32]. Hypoxia can affect electron transfer in the mitochondrial respiratory chain, leading to reduced ATP synthesis. However, ubiquinone can promote ATP synthesis, thereby reducing oxidative stress and protecting cells under hypoxic conditions. The disordered flora can regulate human metabolism through its metabolite tyrosine, which is absorbed into the blood and participates in the metabolic pathways of the human body, such as ubiquinone and other terpenoid-quinone biosynthesis. Through this channel, the content of ubiquinone in the body is increased to promote ATP synthesis, thus reducing oxidative stress and protecting cells under anoxic conditions, helping the human body confront acute altitude hypoxia stress.

In addition, to the best of our knowledge, this is the first study of a relationship between tyrosine and EPO content. Under the high-altitude hypoxia environment, people with hypoxemia present flora disorder and changes in their metabolites as well as increased tyrosine that is absorbed into the blood and affects metabolism and the regulation of EPO production. First, we analyzed the content of EPO in AHAH subjects, finding that it was significantly higher than that of the C group. Then, we analyzed the correlation between EPO and five serum differential metabolites, and the results showed that EPO had a significant positive correlation with tyrosine. Tyrosine demonstrated positive associations with disordered flora such as *Bacteroides*, *Parabacteroides,* and *Odoribacter* while exhibiting a negative correlation with *Saccharibactria_Genera _incertae _Sedis*. These findings indicate that an increase in the relative abundance of *Bacteroides*, *Parabacteroides,* and *Odoribacter*, coupled with a decrease in the relative abundance of *Saccharibactria_Genera _incertae _Sedis*, leads to the increase in its metabolite tyrosine, and the absorption of tyrosine into the blood increases the EPO content of AHAH subjects, helping the body cope with altitude hypoxia stress.

These results provide a new perspective for the study of AHAH. Studying AHAH from the perspective of the intestinal flora will provide a new perspective for its treatment and further elucidate the relationship between intestinal flora, its metabolites, and human physiology. However, this study had some limitations. The sample size was relatively small. In order to strengthen the validity of these findings, further research with a larger sample size is needed. Future research should explore the relationship between intestinal flora, its metabolites, and acute altitude hypoxia and investigate the therapeutic effect of intestinal flora interventions on AHAH.

## Figures and Tables

**Figure 1 microorganisms-11-02284-f001:**
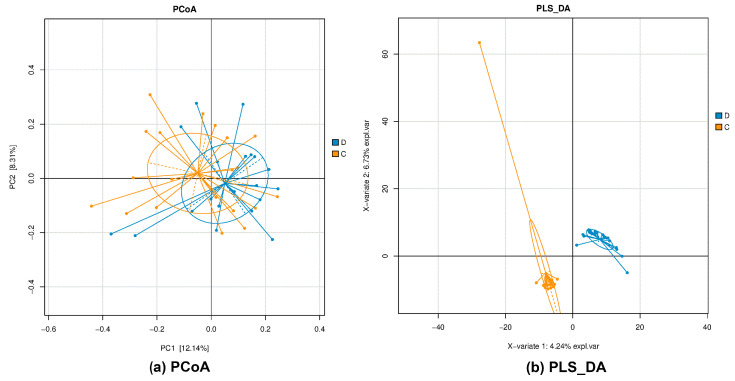
(**a**) PCoA plot, indicating clear differences in the microbial communities between the CD groups; (**b**) PLS-DA plot, indicating significant differences in the bacterial community structure between C and D groups.

**Figure 2 microorganisms-11-02284-f002:**
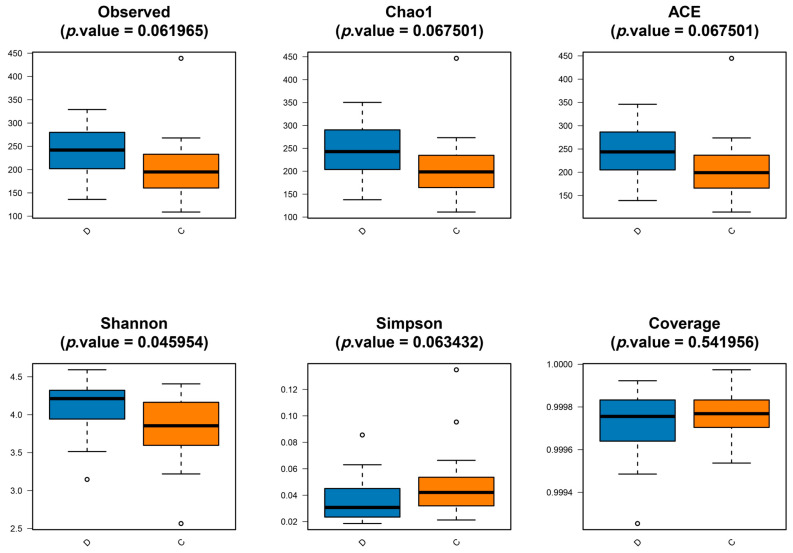
Based on the Wilcoxon rank sum test, difference analysis of α diversity indices such as Observed, Chao, ACE, Shannon, Simpson, and Coverage indices between C and D groups.

**Figure 3 microorganisms-11-02284-f003:**
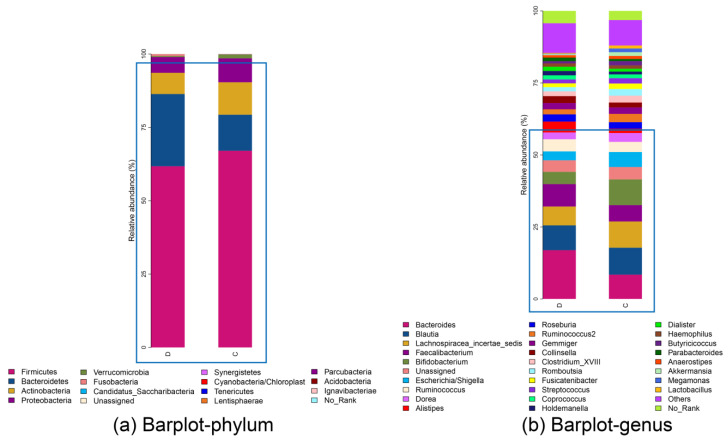
Bar charts of species composition in C and D groups. (**a**) Species composition of multiple samples at the phylum level, with the blue wireframe highlighting the top 4 phyla based on their relative abundance. (**b**) Species composition of multiple samples at the genus level, and the blue wireframe indicates the top 10 genera with relative abundance.

**Figure 4 microorganisms-11-02284-f004:**
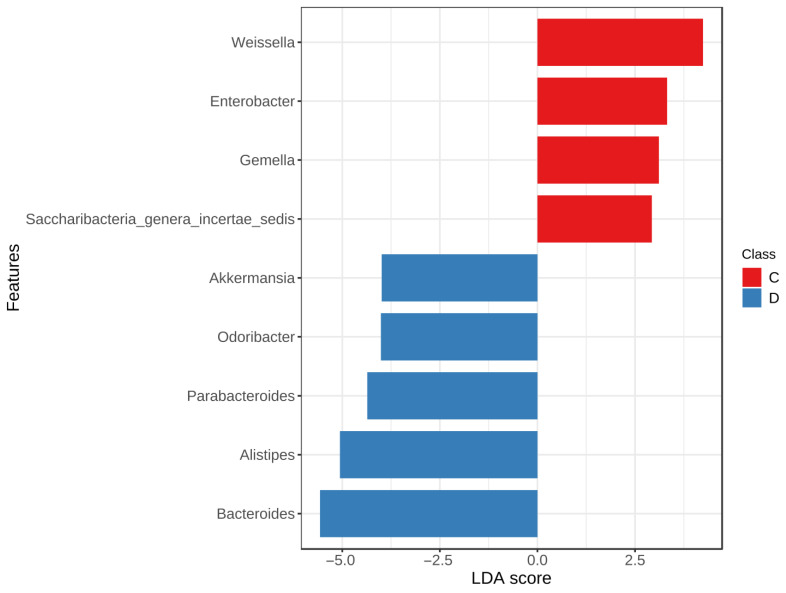
Bar chart based on Lefse, which is used to identify the differentially abundant genera identified between C and D groups.

**Figure 5 microorganisms-11-02284-f005:**
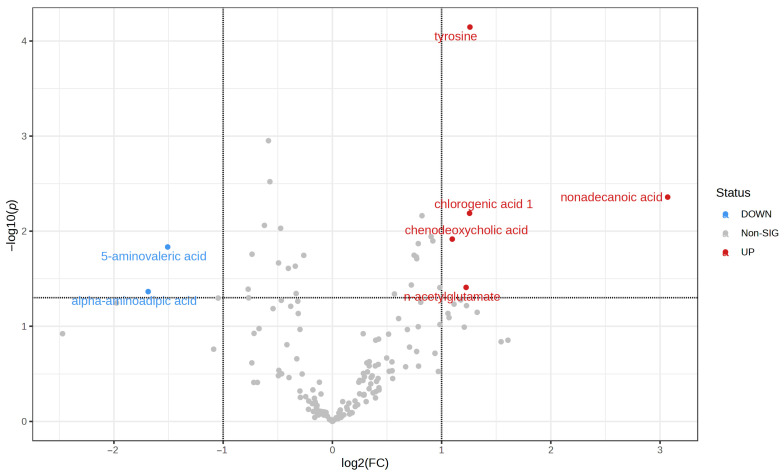
A volcano plot for metabolic findings in stool samples.

**Figure 6 microorganisms-11-02284-f006:**
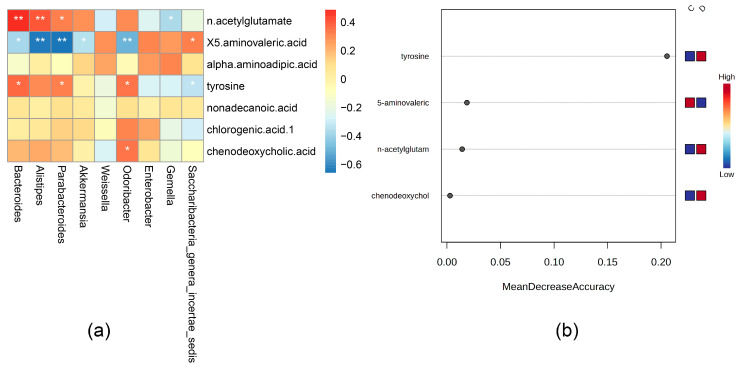
(**a**) Correlation between differential bacterial genera and differential metabolites in feces. (**b**) The random forest method was used to analyze the mean decrease accuracy of metabolites significantly related to intestinal flora in an orderly manner. The larger the value, the more important the variable. Significance: * *p* < 0.05 and ** *p* < 0.01.

**Figure 7 microorganisms-11-02284-f007:**
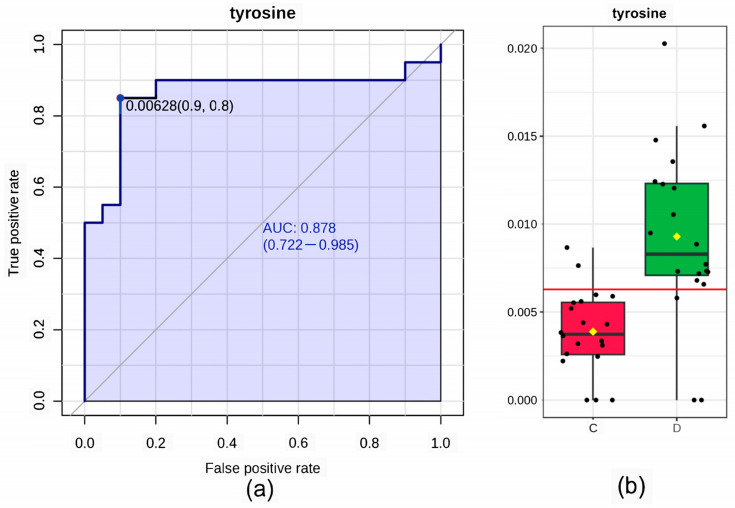
(**a**) In ROC curve analysis, the abscissa indicates specificity of 1, that is, the false positive rate. The closer the value is to zero, the higher the accuracy of the model. The ordinate represents the sensitivity, that is, the true positive rate. The larger the value, the higher the accuracy of the model. (**b**) Box plot of the concentrations of tyrosine between two groups within the dataset. A horizontal line is in red indicating the optimal cutoff.

**Figure 8 microorganisms-11-02284-f008:**
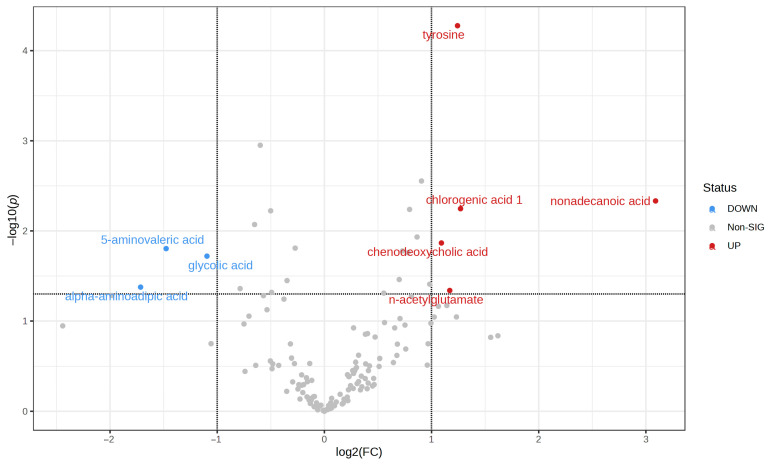
Volcano plot for metabolic findings in serum samples.

**Figure 9 microorganisms-11-02284-f009:**
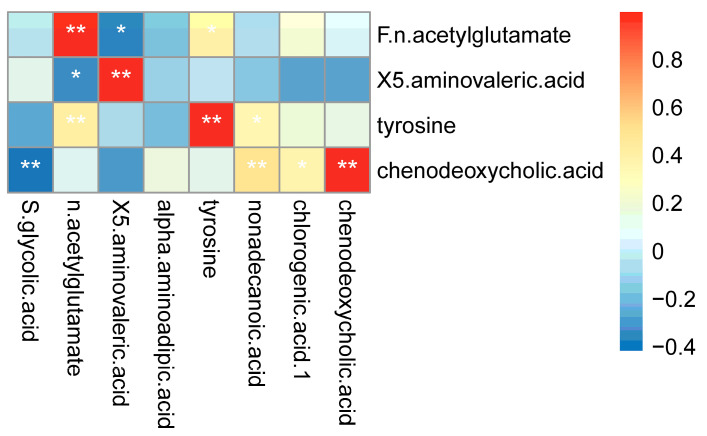
Correlation between different metabolites screened from feces and serum. Significance: * *p* < 0.05 and ** *p* < 0.01.

**Figure 10 microorganisms-11-02284-f010:**
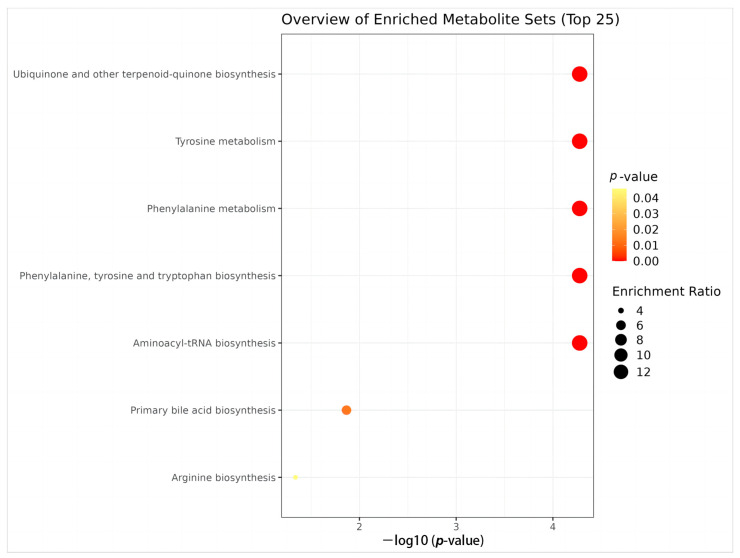
Pathway enrichment analysis results for the 9 significantly different metabolic pathways selected using the KEGG database. The size of the black circle indicates the magnitude of the pathway enrichment rate.

**Figure 11 microorganisms-11-02284-f011:**
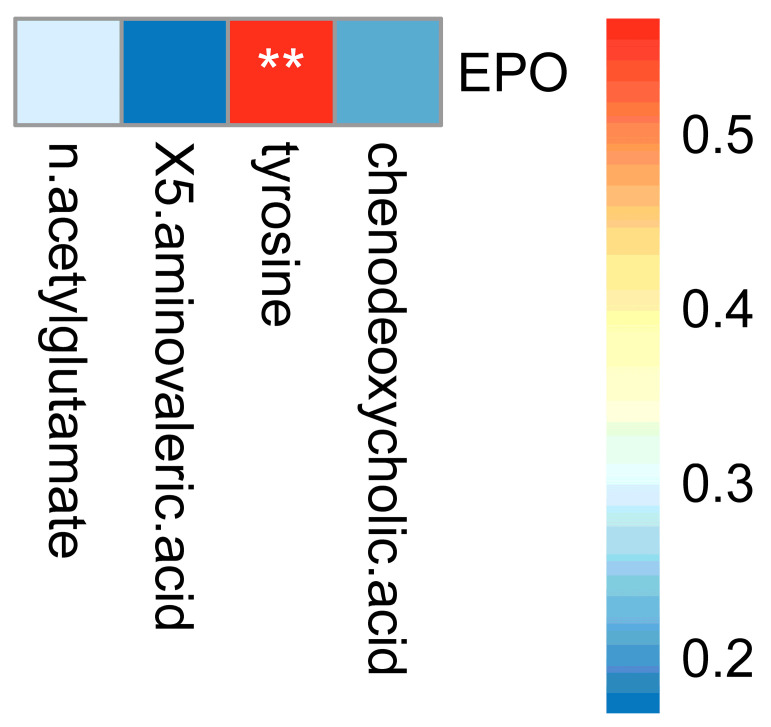
Correlation between screened 4 serum differential metabolites and EPO. Significance: ** *p* < 0.01.

**Table 1 microorganisms-11-02284-t001:** ROC analysis results.

	AUC	*p*-Value	FC	Clusters
Tyrosine	0.878	0.000071348	−1.2594	1
n-acetylglutamate	0.715	0.039040173	−1.2242	3
Chenodeoxycholic acid	0.715	0.012122062	−1.0975	3
5-aminovaleric acid	0.6275	0.014665377	1.508	2

**Table 2 microorganisms-11-02284-t002:** Independent sample *t*-test results of EPO content between C and D groups.

Variable Name	Variable Value	Sample	Mean (mIU/mL)	Standard Deviation (mIU/mL)	t-Value	*p*-Value	Mean Difference	Cohen’s d-Value
EPO	C	20	9.798	5.903	−3.231	0.003 **	9.777	1.022
	D	20	19.575	12.177				
	total	40	14.687	10.664				

Note: ** *p* < 0.01.

## Data Availability

The data used to support the results of this study can be found in the Appendix A.

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
