# Peer review of "Effects of Hypoxemia by Acute High-Altitude Exposure on Human Intestinal Flora and Metabolism"

_microorganisms, 2023, doi:10.3390/microorganisms11092284_

Round 1

Reviewer 1 Report

The paper is very interesting, examining the effects of hypoxemia caused by acute high-altitude hypoxia (AHAH) exposure on human intestinal flora and its metabolites. Authors compared  changes in the intestinal flora, metabolism, and erythropoietin content in the AHAH population under altitude hypoxia conditions using 16S rRNA sequencing, metabonomics, and erythropoietin content. The results showed that the flora and metabolites in  the hypoxemia group were altered. compared with that in the control group. 

In my eyes, It would have been interesting to include also a third arm, constitued by Lanzhou residents not developing symptoms of hypoxemia less than 48 hours after entering Gannan. This could allow controlling changes induced by hypoxia not translating into symptoms but maybe changing the gut microbiota 

Author Response

Dear Reviewer,

Thank you sincerely for your insightful comments and suggestions regarding our manuscript. We deeply appreciate your interest in our study investigating the effects of hypoxemia resulting from acute high-altitude hypoxia on the human intestinal flora and its metabolites. Please find our responses to your suggestions below:

"In my eyes, It would have been interesting to include also a third arm, constituted by Lanzhou residents not developing symptoms of hypoxemia less than 48 hours after entering Gannan. This could allow controlling changes induced by hypoxia not translating into symptoms but maybe changing the gut microbiota."

We acknowledge your suggestion to include a third arm in the study, consisting of Lanzhou residents who did not develop symptoms of hypoxemia within 48 hours of entering Gannan. This additional group would allow us to control for changes induced by hypoxia that may not manifest as symptoms but could potentially affect the gut microbiota. We find this suggestion intriguing. However, it is important to note that our study specifically focused on individuals who experienced hypoxemia after acute high-altitude exposure, with the aim of identifying the effects of hypoxemia resulting from acute high-altitude exposure on the human intestinal flora and metabolism. Therefore, including a group of individuals who did not experience hypoxemia would deviate from the main theme of our research.

We want to emphasize that our research team is dedicated to the field of high-altitude medicine and the study of the gut microbiota. We highly value your suggestions and will continue our research efforts to further explore the relationship between the gut microbiota and high-altitude adaptation. In future studies, we plan to compare individuals from low-altitude regions, those who experienced acute high-altitude exposure without developing hypoxemia, and those who did experience hypoxemia. This comparative approach will allow us to identify the gut microbiota that may play a crucial role in assisting the host's adaptation to acute high-altitude exposure, shedding light on the significant role of gut microbiota and its metabolism in human adaptation to high-altitude environments.

Once again, we express our sincere gratitude for your valuable feedback. Your suggestions have provided us with important insights and considerations for our future research. If you have any further comments or recommendations, we would be grateful to receive them.

Thank you for your time and consideration.

Best regards,

LZ

Reviewer 2 Report

Minor corrections:

1.                Review phrases in lines 340, 357, and 365.

2.                The authors must add a paragraph stating why the experiments were done within 48 hours of the altitude change and discuss the consequences of making the evaluations shorter or longer.

The manuscript is well written.

Author Response

Dear Reviewer,

Thank you for your valuable feedback and insightful suggestions on our manuscript. We appreciate the time and effort you have dedicated to reviewing our work. We have carefully considered your comments and have made the necessary revisions accordingly. Please find our responses to each of your suggestions below:

1 Review phrases in lines 340, 357, and 365:

We sincerely apologize for the oversight in failing to include the specific review phrases mentioned in lines 340, 357, and 365. Line 340 has been revised to include the missing words, line 357 has been amended to remove repetitive statements about AUC, and line 365 has been corrected to address the issue of lower case in the phrases referring to the C and D groups. We have conducted a comprehensive review of our manuscript and have made the required corrections to enhance clarity and coherence in those specific sections.

2 The authors must add a paragraph stating why the experiments were done within 48 hours of the altitude change and discuss the consequences of making the evaluations shorter or longer:

In response to your suggestion, we have added a paragraph to the "Subjects" section, addressing the timing of the experiments. The revised paragraph is as follows:

"Hypoxemia typically manifests within 6 to 12 hours after exposure, reaching its peak se-verity after the initial night [14]. In the absence of additional altitude gain, hypoxemia generally improves within 48 hours [15,16]. Therefore, when investigating the influence of hypoxemia on the intestinal flora, it is important to select individuals who experience hypoxemia within 48 hours of entering high-altitude areas. Selecting a timeframe exceeding 48 hours would likely involve individuals who have entered the high-altitude expo-sure adaptation phase, during which hypoxemia symptoms disappear for most patients."

We believe that this addition provides a comprehensive understanding of the experimental design and strengthens the significance of our findings.

Once again, we would like to express our deep gratitude for your valuable feedback, which has undoubtedly improved the quality of our manuscript. We genuinely appreciate your expertise and guidance throughout this review process. Should you have any further suggestions or concerns, please do not hesitate to let us know.

Kind regards,

LZ

Reviewer 3 Report

Authors have investigated the changes in the intestinal flora and metabolites in AHAH samples using 16S rRNA, metabonomic, and EPO analyses. The result of this study suggests a significant change in the intestinal flora of AHAH subjects. The important conclusion is the positive correlation between tyrosine and EPO in high altitude hypoxia which is deriving the increased blood oxygen demand in the patients. Authors can also investigate chronic hypoxemia to check if the changes are similar to what is seen under acute conditions. Overall it’s a significant work and the results are presented clearly.

Author Response

Dear Reviewer,

Thank you sincerely for your insightful comments and suggestions regarding our manuscript. We deeply appreciate your interest in our study investigating the effects of hypoxemia resulting from acute high-altitude hypoxia on the human intestinal flora and its metabolites. Please find our responses to each of your suggestions below: 

“Authors can also investigate chronic hypoxemia to check if the changes are similar to what is seen under acute conditions. Overall it’s a significant work and the results are presented clearly.”

We greatly appreciate your recognition of our study's investigation into the changes in the intestinal flora and metabolites in individuals with acute high-altitude hypoxia (AHAH) using 16S rRNA, metabonomic, and erythropoietin (EPO) analyses. Our findings indeed suggest a significant alteration in the intestinal flora of individuals with AHAH. We also agree that the positive correlation between tyrosine and EPO in high-altitude hypoxia highlights the increased blood oxygen demand in these patients, emphasizing the physiological responses to hypoxia.

Moreover, we acknowledge the importance of exploring chronic hypoxemia and determining whether the observed changes are similar to those seen under acute conditions. We wholeheartedly agree with your suggestion and will include chronic hypoxemia in our future research endeavors. By investigating the effects of both acute and chronic hypoxemia on the intestinal flora and metabolites, we aim to gain a comprehensive understanding of the adaptive responses of the gut microbiota in different hypoxic conditions.

We want to emphasize that our research team is dedicated to the field of high-altitude medicine, and we deeply value your suggestions. We will continue our research efforts to further explore the relationship between the gut microbiota and high-altitude adaptation. In future articles, we plan to compare the gut microbiota of individuals from low-altitude regions, those with acute high-altitude exposure-induced hypoxemia, and those with long-term high-altitude exposure-induced chronic hypoxemia. This comprehensive approach will enhance our understanding of the gut microbiota's role in different hypoxic conditions.

Once again, we express our sincere gratitude for your valuable feedback. Your suggestions have provided us with important directions for our future research. If you have any further comments or recommendations, we would be grateful to receive them.

Thank you for your time and consideration.

Best regards,

LZ

Reviewer 4 Report

Qi et al. investigated the impact of acute high-altitude hypoxia (AHAH) on human gut microbiota and metabolites. Using 16S rRNA sequencing, metabonomics, and erythropoietin analysis, they comprehensively analyzed changes in flora, metabolism, and erythropoietin content under altitude hypoxia. Compared to the control group (C group), the hypoxemia group (D group) displayed altered flora and metabolites. Tyrosine emerged as a metabolic marker for flora alterations. Fecal and serum analyses revealed microbial metabolites' presence in the blood, influencing human metabolism. Notably, tyrosine correlated significantly with erythropoietin content, indicating the role of gut microbiota and metabolites in regulating altitude stress through erythropoietin. These findings offer insights into adapting to and preventing AHAH. This study performed a comprehensive experiment to show the impact of AHAH on gut flora, although they only included a limited number of samples. I only have some minor comments.

Abstract

Line 23. The full name of EPO should be presented when it was first stated in the manuscript.

Line 59. More information about the subject need to clarify in this section. For example, gender, age, lifestyle, eating habits, etc., because those factors are well-known to impact the gut bacteria.

Line 169. Statistical Analyses. Only Shannon, Simpson, and Chao1 indexes were analyzed in this study? but the authors showed Observed, Chao, ACE, Shannon, Simpson, and Coverage indexes, how to calculate the rest of indexes?

Figure 1 should be listed as a supplementary information.

The p-value of Observed, Chao, ACE is same in Figure 2b? what method did the authors use?

Beta-diversity normally uses pcoa plot, Did the authors have pcoa plot to show the composition difference?

Line 264. Phyla level or family level? It would be great to show the top 10 or top 15 phyla or genera level in figure3.

Author Response

Dear Reviewer,

Thank you for your valuable feedback and suggestions regarding our manuscript titled "Effects of hypoxemia by acute high-altitude exposure on human intestinal flora and metabolism." We appreciate your thorough review and have addressed each of your comments below:

Line 23: The full name of EPO should be presented when it was first stated in the manuscript.

Response: Thank you for pointing that out. We have made the necessary correction in the Abstract section, where we first mention erythropoietin (EPO), providing its full name.

Line 59: More information about the subjects needs to be clarified in this section. For example, gender, age, lifestyle, eating habits, etc., because those factors are well-known to impact gut bacteria.

Response: We have added additional information about the participants' daily diet in Supplementary Table 1. This table includes data on Grain Intake (g/d), Tuber Crop Intake (g/d), Vegetable Intake (g/d), Fruit Intake (g/d), Meat Intake (g/d), Egg Intake (g/d), Milk and Dairy Products Intake (ml/d), Fat Intake (g/d), and Water Intake (ml/d). Since this study is essentially a self-controlled experiment, analyzing the changes in intestinal flora and metabolism of the same individual before and after acute high altitude exposure, its characteristic is that it can reduce the influence of confounding factors such as diet. Therefore, we have included this diet-related data in the baseline data section but have not included it in the main body of the manuscript.

Line 169: Statistical Analyses. Only Shannon, Simpson, and Chao1 indexes were analyzed in this study? But the authors showed Observed, Chao, ACE, Shannon, Simpson, and Coverage indexes. How to calculate the rest of the indexes?

Response: We apologize for the confusion. We have provided additional clarification in section 2.6. Statistical Analyses. For alpha diversity analysis, we used the vegan package (v2.5.6) in R to calculate diversity indices, including Observed, Chao1, ACE, Shannon, Simpson, and Coverage. Additionally, we employed R v3.5.1 for the statistical analysis of the diversity indices.

Figure 1 should be listed as supplementary information.

Response: Thank you for your suggestion. We have moved Figure 1 to the supplementary materials and renamed it as "Supplementary Figure S1."

The p-value of Observed, Chao, ACE is the same in Figure 2b? What method did the authors use?

Response: We performed Wilcoxon rank-sum tests to analyze the differences in alpha diversity indices between the CD groups. The Wilcoxon test is a non-parametric method that compares the ranks of the data rather than the actual values. The ranks refer to the positions of the data when sorted in ascending order. For example, if we have a set of data {3, 1, 4, 2}, the sorted order would be {1, 2, 3, 4}, and the corresponding ranks would be {2, 4, 1, 3}. In the Wilcoxon test, we compare the ranks of two groups of data to determine if there is a significant difference between them. Although the Chao1 and ACE indices have different numerical values for each sample, they share the same ranks, resulting in the same p-values. Additionally, we reanalyzed the data using the Benjamini-Hochberg Adjusted P value online tool, and the results were consistent with our previous analysis. To provide you with a more comprehensive understanding, we have added Supplementary Table S3, which includes the raw data of the Alpha Diversity Indices of the C and D groups.

Beta-diversity normally uses PCoA plot. Did the authors have a PCoA plot to show the composition difference?

Response: Thank you for your suggestion. We have included a Principal Coordinate Analysis (PCoA) plot in Figure 1, which demonstrates the differences in microbial community composition between the CD groups.

Line 264: Phyla level or family level? It would be great to show the top 10 or top 15 phyla or genera levels in Figure 3.

Response: We have replaced the original content with a bar plot illustrating the species composition at the phylum level. Furthermore, we have made necessary adjustments to the graphics and their corresponding descriptions. In addition, we have emphasized the four most abundant phyla and highlighted the top ten genera in the figure using blue boxes.

Once again, we sincerely appreciate your valuable feedback and suggestions. We believe that the modifications we have made address your concerns and have improved the quality of our manuscript. Please feel free to reach out if you have any further comments or suggestions.

Thank you.

Best regards,

LZ

Round 2

Reviewer 1 Report

To me the paper now is fine.